# FUNCTIONAL-LEVEL UNCERTAINTY QUANTIFICATION FOR CALIBRATED FINE-TUNING ON LLMS

## ABSTRACT

Accurate uncertainty quantification in large language models (LLMs) is essential for providing credible confidence estimates over their outputs. However, fine-tuned LLMs often exhibit overconfidence in uncertain predictions, which stems from their limited ability to generalize with sparse data. Existing parameter efficient fine-tuning (PEFT) uncertainty quantification methods for LLMs focus on post fine-tuning stage, and thus fail to address the core issue: limited specialization of PEFT adapters to accurately capture task-specific input-output relationships. To address these limitations, we propose Functional-Level Uncertainty Quantification for Calibrated Fine-Tuning (UQ4CT), which captures and calibrates uncertainty over the space of functions that map input prompts to outputs. We implement UQ4CT during the fine-tuning stage via a mixture-of-experts framework that hierarchically decomposes the functional space. Empirically, UQ4CT achieves over 25% reduction in Expected Calibration Error (ECE) while preserving high accuracy across five benchmarks. Even under distribution shift, UQ4CT maintains superior ECE performance with high accuracy, showcasing improved generalizability.

## 1 INTRODUCTION

Quantifying the credibility of outputs has been one of the most important problems around large language models (LLMs)(Chang et al., 2024). In particular, fine-tuned LLMs often struggle with overconfidence in their outputs due to limited training data, failing to reflect the true credibility of their answers(Xiao et al., 2022; He et al., 2023; Tian et al., 2023; OpenAI, 2023). Such overconfidence can assert misinformation with high certainty, making it difficult for users to discern truth from falsehood. This has become a crucial problem in safety-critical decision making and scientific domains where data is relatively limited, such as formal proof generation, climate science, and healthcare (Singhal et al., 2022; Wu et al., 2023a; Lampinen et al., 2023; Li et al., 2022). Methods that enhance uncertainty quantification for fine-tuned LLMs are therefore essential to ensure trustworthy predictions.

One salient challenge of uncertainty quantification in large language models is the trade-off among accuracy, calibration, and efficiency. Ideally, one seeks to calibrate model uncertainty without degrading accuracy or slowing output generation. Recent approaches often focus on prompt perturbation—modifying the model input and quantifying the resulting prediction variance (Hou et al., 2023; Gao et al., 2024)—or sampling multiple completions to measure prediction disagreement (Farquhar et al., 2024). However, these methods generally assume the model is already well-aligned with the data distribution, and thus struggle to capture uncertainty arising during fine-tuning, especially the generalization gap due to adaptation on limited data. Additionally, since these methods require multiple forward passes per input, they incur significant computational overhead, limiting scalability.

Beyond prompt-level approaches, Bayesian methods and ensemble-based uncertainty quantification have been established for fine-tuned LLMs, often in conjunction with low-rank adaptation (LoRA) (Hu et al., 2021a). Methods such as Monte Carlo dropout (Gal & Ghahramani, 2016), checkpoint ensembles (Chen et al., 2017), deep ensembles (Lakshminarayanan et al., 2017; Wang et al., 2023; Zhai et al., 2023), and Laplace-LoRA (Yang et al., 2024a) apply Bayesian inference or ensembling over LoRA parameters to capture uncertainty arising from model adaptation and limited data. While Bayesian and ensemble methods estimate uncertainty after fine-tuning by analyzing the learned parameter space, they do not address the limitations caused by sparse data during fine-tuning. This post hoc perspective can miss uncertainty that arises when adapting to new tasks with limited data. Recent

work has also explored information-theoretic evidential calibration for LLMs, such as evidential deep learning approaches that directly model predictive uncertainty and provide information-theoretic guarantees (Li et al., 2025).

To overcome this, we shift focus from parameter-space to functional-space uncertainty quantification. The functional space encompasses the input-output mappings the model can realize, capturing the true variability in its predictions. By calibrating uncertainty at this level during fine-tuning, we ensure the model's confidence better reflects its actual predictive reliability.

We therefore introduce Functional-Level Uncertainty Quantification for Calibrated Fine-Tuning (UQ4CT), a method that explicitly calibrates functional-level uncertainty in LLMs during fine-tuning. UQ4CT leverages ensembles of LoRA modules at each layer to construct a rich set of basis functions. We then employ a Mixture-of-Experts (MoE) architecture (Li et al., 2024) to hierarchically combine these basis functions, forming a flexible functional space (see Figure 1). During fine-tuning, UQ4CT jointly learns the LoRA expert parameters and calibrates the prompt-dependent function mixture to align functional-level uncertainty with predictive correctness, enabling the model to output calibrated distributions over the functional space.

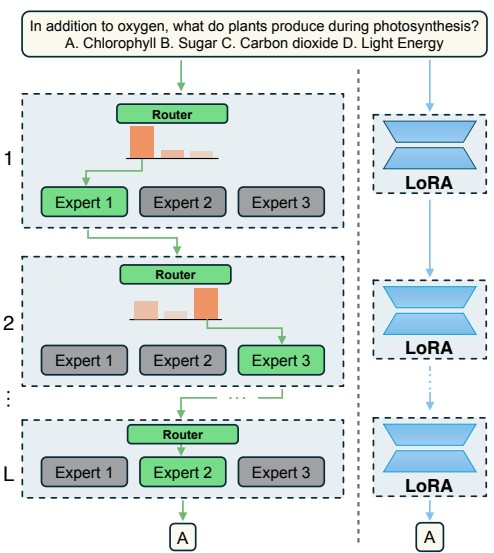

Figure 1: Left: The Mixture of Experts (MoE) architecture captures diverse functional relationships by dynamically routing inputs to different expert modules based on the prompt. Right: The standard LoRA approach lacks such functional-space diversity, limiting its capacity to capture variations in the functional relationships during fine-tuning.

During inference, LoRA experts offer diverse functional relationships acquired during fine-tuning, while MoE routers dynamically select the most relevant experts for each input prompt. This selection is guided by functional-level uncertainty calibration performed throughout fine-tuning, which aims to optimize the choice of the correct functional relationship for each prompt. More accurate expert selection enables the model to learn diverse functional relationships. As a result, the model's uncertainty estimates become better aligned, enhancing calibration without compromising accuracy. To summarize, our contributions include:

- A novel uncertainty quantification approach for LLMs with MoE architecture during fine-tuning to quantify functional-level uncertainty and align with the probability of predictive correctness, which mitigates overconfidence issue and improves generalizability.

- A new calibration loss that incorporates predictive correctness probability to dynamically align the prompt-dependent LoRA mixture for better uncertainty estimation.

- Hierarchical decomposition of functional-level uncertainty into layer-wise mixture weights with guarantee that our calibration loss aligns mixture weights with predictive correctness.

- More than 25% expected calibration error (ECE) reduction on 4 common-sense reasoning tasks and 1 domain-specific question answering task; improved ECE performance without compromising accuracy *under distribution shift* on 2 common-sense reasoning tasks and 4 domain-specific question answering tasks.

## 2 PRELIMINARIES

**Low-rank Adaptation (LoRA).** LLMs have numerous large weight matrices to perform matrix multiplication, denoted as $\mathbf{W}_0 \in \mathbb{R}^{n_{out} \times n_{in}}$ that maps inputs $\mathbf{x}$ to outputs $\mathbf{h}$. Hu et al. (2021a) proposes LoRA, which fixes $\mathbf{W}_0$ and introduces a low-rank perturbation $\Delta\mathbf{W}$ to the weight matrix:

$$\mathbf{h} = \mathbf{W}_0\mathbf{x} + \Delta\mathbf{W}\mathbf{x} = \mathbf{W}_0\mathbf{x} + \mathbf{B}\mathbf{A}\mathbf{x}. \tag{1}$$

Here, $\Delta\mathbf{W}$ is calculated as the product of two matrices, $\mathbf{B} \in \mathbb{R}^{n_{\text{out}} \times n_{\text{lr}}}$ and $\mathbf{A} \in \mathbb{R}^{n_{\text{lr}} \times n_{\text{in}}}$ where $n_{\text{lr}}$ is significantly smaller than $n_{\text{in}}$ or $n_{\text{out}}$. For example, we use $n_{\text{lr}} = 32$ while $n_{\text{in}} = n_{\text{out}} = 4096$ for the Llama3.1-8B model (Dubey et al., 2024). Therefore, the total number of LoRA parameters for this $\Delta\mathbf{W}$ is $n_{\text{lr}}(n_{\text{in}} + n_{\text{out}})$, which is far smaller than the parameter count of the full matrix, $n_{\text{in}}n_{\text{out}}$. One of the key motivations of incorporating LoRA to fine-tune LLMs is the vast amount of memory cost reduction compared with fine-tuning on the full model. For an LLM with 7 billion parameters, maintaining the average gradient and average squared gradients for optimization multiplies the memory required by a factor of 3 compared to simply loading model weights. LoRA greatly mitigates this memory cost as the tripled memory consumption only applies to LoRA adapters.

**Mixture of Experts (MoE).** LoRA Mixture-of-Experts (Li et al., 2024; Wu et al., 2024b) is an efficient approach to scale the number of parameters while maintaining the same computational bounds. LoRA MoE utilizes the top-k router to assign each token to the LoRA experts (Lepikhin et al., 2020). The router is a linear layer that maps the input hidden state $\mathbf{h}$ to a probability distribution of candidate experts.

The plain transformer block in a large language model consists of the $q, k, v$ encoding layers ($\text{FFN}_{q,k,v}$), layer norm (LN) and the feedforward layer (FFN), together with residual connections. Formally, given $\mathbf{h}^1$ as the tokenized input text, the output of $\ell$-th layer is generated as:

$$\mathbf{z}^\ell = f_{attn}(\text{FFN}_{q,k,v}(\text{LN}(\mathbf{h}^{\ell-1}))) + \mathbf{h}^{\ell-1}, \quad \mathbf{h}^\ell = \text{FFN}(\text{LN}(\mathbf{z}^\ell)) + \mathbf{z}^\ell. \tag{2}$$

Here, $f_{attn}$ represents the attention calculation step.

Let $\mathbf{h}^\ell \in \mathbb{R}^{1 \times d}$ $(1 \leq \ell \leq L)$ denote the output hidden state at the $\ell$-th layer of the LLM, where $L$ is the number of LLM layers and $d$ is the hidden dimension. With $\mathbf{W}_r^\ell$ as the trainable router weight at layer $\ell$, the top-k gate router $\tilde{R}(\cdot)$ chooses $k$ experts with highest probability given a hidden state $\mathbf{h}^\ell$:

$$\tilde{R}^\ell(\mathbf{h}^\ell) = \text{Keep-Top-k}(\text{Softmax}(\mathbf{W}_r^\ell \cdot \mathbf{h}^\ell)). \tag{3}$$

Finally, we obtain the final MixLoRA prediction with:

$$\text{MixLoRA}(\mathbf{h}^\ell) = \sum_{k=1}^{K} \tilde{R}^\ell(\mathbf{h}^\ell)_k E_k^\ell(\mathbf{h}^\ell), \quad E_k^\ell(\mathbf{h}^\ell) = \mathbf{W}_{pre}^\ell \cdot \mathbf{h}^\ell + \mathbf{B}_k^\ell \mathbf{A}_k^\ell \cdot \mathbf{h}^\ell \tag{4}$$

where $\mathbf{W}_{pre}^\ell$ is the frozen pretrained weight at layer $\ell$ and $\mathbf{B}_k^\ell \mathbf{A}_k^\ell$ is the k-th LoRA expert.

With MixLoRA defined in Equation 4, we can apply MixLoRA layers at $q, k, v$ encoding and FFN layers:

$$\mathbf{z}^\ell = f_{attn}(\text{MixLoRA}_{q,k,v}(\text{LN}(\mathbf{h}^{\ell-1}))) + \mathbf{h}^{\ell-1}, \quad \mathbf{h}^\ell = \text{MixLoRA}(\text{LN}(\mathbf{z}^\ell)) + \mathbf{z}^\ell. \tag{5}$$

## 3 METHODOLOGY

The high-level goal of UQ4CT is to leverage the ensemble of prompt-dependent LoRA mixture-of-experts (MoE) to guide and calibrate the confidence of the model during fine-tuning. By quantifying the variability in how different LoRA experts are combined for each input, UQ4CT enables the model to adaptively select expert mixtures that reflect the true uncertainty in its predictions. Our approach not only encourages the model to exploit confident expert combinations for accurate predictions but also promotes exploration of alternative experts when uncertainty is high, ultimately leading to better-calibrated and more reliable model outputs.

### 3.1 DECOMPOSITION OF THE FUNCTIONAL SPACE

Given the immense size of both the pre-training dataset and the model, we posit that the pretrained network contains submodules capable of expressing a wide range of functional relationships present in the data. During fine-tuning, our focus is on the functional space spanned by the model, which can be effectively captured as a mixture of LoRA experts, each representing a distinct basis function. However, naively decomposing the full functional space by considering all possible combinations of these experts quickly leads to a combinatorial explosion.

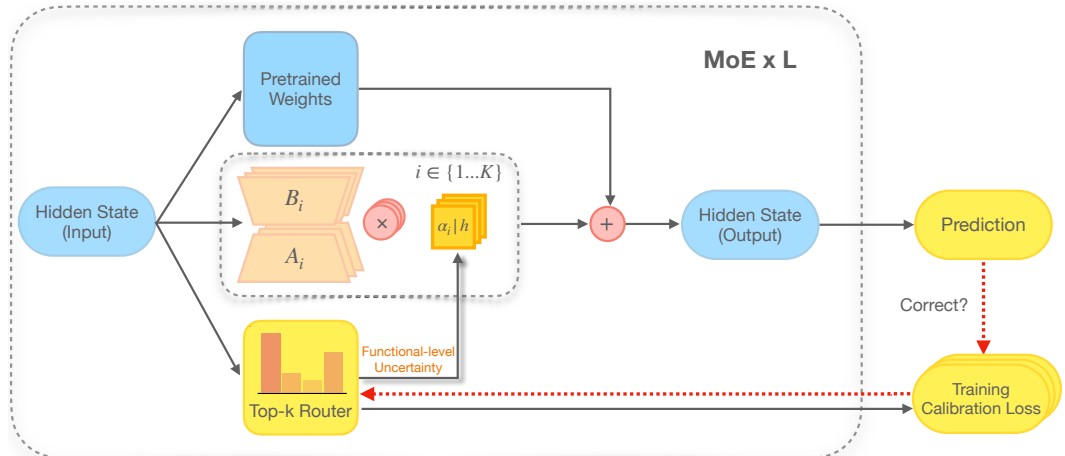

Figure 2: MoE architecture to capture functional-level uncertainty. LoRA experts $(B_i, A_i)$ capture diverse functional bases, while the top-$k$ router assigns mixture weights based on the input hidden state. The calibration loss aligns functional-level uncertainty with prediction correctness, encouraging confident expert selection for correct predictions and higher uncertainty for incorrect ones.

**Naive Decomposition.** Denote the input prompt as $\mathbf{x}$ and the functional map from input to output as $f$. A straightforward approach is to express the function as a sum over all possible compositions of $K$ submodules per layer, across $L$ layers:

$$f(\mathbf{x}) = \sum_{k^1, \cdots, k^L = 1}^{K} \alpha_{k^1, \ldots, k^L} g_{k^L}^L \left( \cdots \left( g_{k^l}^l \left( \ldots g_{k^1}^1(\mathbf{x}) \right) \right) \right), \tag{6}$$

where each $g_{k^\ell}^\ell$ denotes a particular variant (e.g., LoRA-adapted) of the $\ell$-th block and

$$g_{k^\ell}^\ell(\mathbf{h}^\ell) = \mathbf{h}^\ell + E_{k^\ell}^\ell \left( f_{trans}^\ell(\mathbf{h}^\ell) \right), \tag{7}$$

here $f_{trans}^\ell$ represents the necessary non-fine-tuned operations within a transformer block (i.e. layer norm, attention calculation, etc.) and as defined in Eq. 4, $E_{k^\ell}^\ell$ denotes the parameterized adaptation associated with the $k^\ell$-th expert in the $\ell$-th layer.

However, this naive decomposition is intractable in practice, as it requires keeping track of $K^L$ mixture weights $\alpha_{k^1, \ldots, k^L}$. This is an exponential growth in the number of parameters with respect to both the number of layers $L$ and the number of submodules per layer $K$, which makes the direct approach computationally infeasible for realistic network sizes.

**Hierarchical Decomposition.** To address the combinatorial explosion of mixture weights, we instead propose a hierarchical decomposition. Here, the mixture at each layer is formed independently, and the output of each layer is a weighted sum over its submodules, with the weights themselves being layer-specific:

$$f(\mathbf{x}) = \sum_{k^L = 1}^{K} \alpha_{k^L}^L g_{k^L}^L \left( \cdots \left( \sum_{k^l = 1}^{K} \alpha_{k^l}^l g_{k^l}^l \left( \cdots \sum_{k^1 = 1}^{K} \alpha_{k^1}^1 g_{k^1}^1(\mathbf{x}) \right) \right) \right). \tag{8}$$

Instead of needing $K^L$ mixture weights, this hierarchical structure only requires $K \cdot L$ weights $\alpha_{k^\ell}^\ell$, one for each submodule in each layer. This dramatically reduces the parameters required and makes the decomposition tractable, while still enabling a rich set of compositional functions.

**Dynamic, Input-Dependent Routing.** To further enhance expressivity and efficiency, we allow the mixture weights to depend dynamically on the input at each layer. Specifically, we set the mixture weights to be a sparse routing function $R^\ell$ of the hidden state $\mathbf{h}^\ell$ at each layer, where $\alpha_{k^\ell}^\ell = \alpha_{k^\ell}^\ell(\mathbf{h}^\ell) = R_{k^\ell}^\ell(f_{trans}^\ell(\mathbf{h}^\ell))$.

Substituting this definition into the hierarchical mixture, the overall function $f(\mathbf{x})$ becomes:

$$f(\mathbf{x}) = \sum_{k^L = 1}^{K} R_{k^L}^L(f_{trans}^L(\mathbf{h}^L)) g_{k^L}^L \left( \cdots \left( \sum_{k^1 = 1}^{K} R_{k^1}^1(f_{trans}^1(\mathbf{h}^1)) g_{k^1}^1(\mathbf{x}) \right) \right). \tag{9}$$

This recursive formulation shows that at each layer, the submodules are weighted according to the input-dependent routing function, which adapts based on the current hidden state.

For a layer-wise perspective, the computation at each layer $\ell$ can be explicitly written as:

$$\mathbf{h}^{\ell+1} = \sum_{k^\ell=1}^{K} R_{k^\ell}^\ell(f_{trans}^\ell(\mathbf{h}^\ell))g_{k^\ell}^\ell(\mathbf{h}^\ell). \tag{10}$$

Here, $R^\ell$ selects the most relevant submodules for a given input, allowing the network to adaptively compose its computation path at each layer in a sparse fashion.

## 3.2 QUANTIFYING FLC WITH MIXLORA

In the previous section, we have established a parsimonious representation of the functional space. In this section, we choose a simple function to encode the functional level uncertainty and provide the intuition as follows. We first note that given a fixed MoE model architecture, the larger weight a mixture component has, the more certain we are about that component.

The model uncertainty of the MixLoRA architecture is quantified by considering perturbations $\Delta f(\mathbf{x})$ to the model $f(\mathbf{x})$. Following the discussion and notation in Sec. 3.1, we can show that these perturbations are instantiated in the space of the mixture weights $\alpha$ (as defined in Eq. 8 and 9):

**Fact 3.1** (Model Perturbation Structure). *Under regularity assumptions on the residual connection architecture, perturbations $\Delta f(x)$ to the model $f(x)$ approximately decompose as:*

$$\Delta f(x) \approx \sum_{\ell=1}^{L} \sum_{k^\ell=1}^{K} \Delta\alpha_{k^\ell}^\ell(\mathbf{h}^\ell) \cdot g_{k^\ell}^\ell(\mathbf{h}^\ell).$$

In high-dimensional settings, the basis functions are approximately orthogonal. Thus, the perturbation $\Delta f(x)$ is entirely represented by the set $\left\{\Delta\alpha_{k^\ell}^\ell(\mathbf{h}^\ell)\right\}_{k^\ell=1,...,K}^{\ell=1,...,L}$. Therefore, the functional-level confidence (FLC) can be generally modeled as a linear function over the mixture weights:

$$\text{FLC}(x) = U\left(\left\{\alpha_{k^\ell}^\ell(\mathbf{h}^\ell)\right\}_{k^\ell=1,...,K}^{\ell=1,...,L}\right),$$

where $U(\cdot)$ denotes a linear aggregation function. Details of derivation is presented in Appendix A.1.

In practice, as illustrated in Figure 2, the top-$k$ router at each layer produces a sparse probability vector $\alpha^\ell = (\alpha_1^\ell, ..., \alpha_k^\ell)$, dynamically mixing the basis functions captured by the LoRA experts given the current hidden state. The values of the top-$K$ routing weights that contribute to the final output hidden state serve as a direct quantification of the model's uncertainty at functional-level. We follow the routing mechanisms used in MoE layers (see Eq. (3) and (4)), employing top-2 gate routers for the mixture. At each layer $\ell$, we compute the raw router probabilities and retain the largest two:

$$\tilde{R}^\ell(\mathbf{h}^\ell) = \text{Keep-Top-2}(\text{Softmax}(\mathbf{W}_r^\ell \cdot \mathbf{h}^\ell)). \tag{11}$$

Given an input prompt $x$ of length $s$, we aggregate the router weights over the selected experts and across all layers to estimate the FLC regarding the predicted next token:

$$\text{FLC}(x) = \frac{1}{L} \sum_{\ell=1}^{L} \sum_{i=1}^{2} \tilde{R}_i^\ell(\mathbf{h}^\ell). \tag{12}$$

This formulation provides an efficient approach to quantify functional-level uncertainty in LoRA MoE architectures.

## 3.3 CALIBRATION LOSS

The FLC model provides a principled way to calibrate the mixture parameters against predictive accuracy, enabling better alignment between output distributions and true model confidence. Specifically, for MoE top-$k$ routers, we design the following calibration loss for training:

$$\mathcal{L}_{\text{cal}} = \left(\mathbb{1}\{\text{MixLoRA}(x) = y^*\} - \text{FLC}(x)\right)^2. \tag{13}$$

The first term is an indicator function that equals $1$ if the model prediction matches the ground truth $y^*$ for prompt $x$, and $0$ otherwise, corresponding to a one-hot definition of ground truth confidence.

This calibration loss directly encourages the functional-level uncertainty (FLC) to reflect the true correctness of the model's predictions. As shown in Figure 2, when the mixture model predicts correctly, the loss pushes FLC toward $1$ (high confidence); when incorrect, toward $0$ (low confidence). In other words, when optimized over the data distribution, the calibration loss pushes FLC to represent the probability of prediction correctness. We formally state this property as follows and present the proof in Appendix A.1:

**Proposition 3.2** (Truthfulness of Calibration Loss). *Let the calibration risk be defined as the expectation of the calibration loss over the data distribution:*

$$\bar{\mathcal{L}}_{\text{cal}} = \mathbb{E}_{(x,y^*)\sim\mathcal{D}} \left( \mathbb{1}\{\text{MixLoRA}(x) = y^*\} - \text{FLC}(x) \right)^2 .$$

*If this calibration risk is optimized over the data distribution $\mathcal{D}$, then the optimal solution is $\text{FLC}(x) = \mathbb{P}(\text{MixLoRA}(x) = y^*(x))$; that is, the optimally trained FLC corresponds to the probability that the model's prediction is correct.*

### 3.4 Fine-tuning with the Total Loss

Our proposed calibration loss $\mathcal{L}_{\text{cal}}$ improves predictive reliability by adaptively balancing expert exploitation and exploration according to functional-level uncertainty. As shown in Figure 2, $\mathcal{L}_{\text{cal}}$ aligns uncertainty with predictive correctness, increasing the router probability of the selected expert for correct predictions (exploitation) and decreasing it for incorrect ones (exploration).

Ideally, when the $K$ LoRA experts collectively capture the relevant functional relationships in the data during fine-tuning via cross-entropy loss, $\mathcal{L}_{\text{cal}}$ further guides the model to select appropriate mixtures of LoRA experts conditioned on the input $\mathbf{x}$. This targeted selection enables the model to match the data distribution more closely and provides more calibrated uncertainty estimates.

To ensure balanced expert utilization, we incorporate a load balancing loss $\mathcal{L}_b$ as proposed by Li et al. (2024). Our overall loss function is:

$$\mathcal{L} = \mathbf{CE} + \gamma \cdot \mathcal{L}_b + \beta \cdot \mathcal{L}_{\text{cal}}, \tag{14}$$

where $\mathbf{CE}$ is the cross-entropy loss, and $\gamma, \beta$ are hyperparameters for the auxiliary terms. We fix $\gamma, \beta$ to $1$ for our experiments. Details of $\mathcal{L}_b$ are provided in Appendix A.2.

## 4 Related Work

**Mixture of LoRA Experts.** Large Language Models (LLMs) have achieved impressive performance across diverse NLP tasks (Brown et al., 2020; Hoffmann et al., 2022; Touvron et al., 2023a;d), with instruction fine-tuning (Chung et al., 2022; Iyer et al., 2022; Zheng et al., 2024) further boosting their adaptability for conversational AI (Wu et al., 2023b; Achiam et al., 2023). However, scaling LLMs increases the resource demands of full fine-tuning. Parameter-efficient fine-tuning (PEFT) methods (Mangrulkar et al., 2022)—such as LoRA (Hu et al., 2021b) and its variants (Kopiczko et al., 2023; Hyeon-Woo et al., 2021; Renduchintala et al., 2023; Zhang et al., 2023; Liu et al., 2024)—reduce adaptation costs by updating a subset of parameters.

Recent advances combine PEFT with the Mixture-of-Experts (MoE) framework (Jacobs et al., 1991; Wang et al., 2020), which sparsely activates expert subnetworks for greater model capacity and specialization. MoE-based LLMs leverage expert routing and parameter-efficient adaptations to target new domains or tasks efficiently. Notably, methods such as MoRAL (Yang et al., 2024b), LoRAMoE (Dou et al., 2024), PESC (Wu et al., 2024a), MoE-LoRA (Luo et al., 2024), and MixLoRA (Li et al., 2024) optimize domain-specific routing, mitigate forgetting, and enable scalable, high-throughput training and inference with mixtures of LoRA experts.

**Uncertainty Quantification in LLMs.** Established uncertainty quantification methods have been studied in conjunction with the LoRA structure for LLMs. Monte-Carlo dropout (Gal & Ghahramani, 2016) interprets dropout in neural networks as approximate Bayesian inference in deep Gaussian processes, allowing uncertainty estimates to be obtained from existing LoRA adapters without

Table 1: Performance comparison of different methods fine-tuned with Llama3.1-8B across four common sense reasoning tasks and a domain-specific task. UQ4CT shows substantial ECE improvements while maintaining high accuracy.

| Metrics | Methods | BoolQ | ARC-E | ARC-C | OBQA | ClimateQA |
|---------|---------|-------|-------|-------|------|-----------|
| **ACC ↑** | Base Model | 74.73 | 87.27 | 74.32 | 72.80 | 68.64 |
| | LoRA | $89.73_{0.58}$ | $88.82_{1.82}$ | $78.21_{0.73}$ | $88.00_{1.22}$ | $78.25_{1.29}$ |
| | MC Drop | $89.65_{0.55}$ | $88.16_{1.75}$ | $77.14_{0.69}$ | $87.12_{1.18}$ | $78.19_{1.35}$ |
| | Ensemble | $\mathbf{89.87_{0.43}}$ | $\mathbf{89.14_{1.31}}$ | $78.81_{0.96}$ | $86.47_{0.42}$ | $78.53_{2.98}$ |
| | MixLoRA | $88.68_{0.82}$ | $87.74_{0.36}$ | $78.56_{1.87}$ | $88.27_{0.50}$ | $79.94_{1.29}$ |
| | LA | $89.58_{0.19}$ | $86.22_{3.52}$ | $78.00_{3.76}$ | $86.00_{6.01}$ | $79.82_{3.48}$ |
| | BLoB(Mean) | $89.02_{0.93}$ | $88.71_{0.82}$ | $79.37_{0.71}$ | $87.60_{1.04}$ | $79.02_{0.50}$ |
| | BLoB(N=10) | $89.39_{1.13}$ | $87.96_{0.62}$ | $\mathbf{80.08_{1.55}}$ | $87.13_{0.88}$ | $79.02_{0.50}$ |
| | UQ4CT | $89.17_{1.33}$ | $88.66_{0.20}$ | $79.60_{1.21}$ | $\mathbf{88.40_{0.35}}$ | $\mathbf{79.97_{0.85}}$ |
| **ECE ↓** | Base Model | 6.94 | 13.76 | 11.30 | 11.39 | 14.58 |
| | LoRA | $15.82_{0.57}$ | $6.55_{1.70}$ | $14.07_{0.68}$ | $7.30_{0.43}$ | $13.70_{1.50}$ |
| | MC Drop | $14.73_{0.54}$ | $6.48_{1.74}$ | $14.12_{0.71}$ | $7.24_{0.39}$ | $13.11_{1.46}$ |
| | Ensemble | $14.56_{0.55}$ | $7.08_{0.73}$ | $13.71_{1.29}$ | $8.63_{0.38}$ | $14.69_{0.84}$ |
| | MixLoRA | $15.85_{0.76}$ | $7.79_{0.45}$ | $13.71_{1.90}$ | $6.58_{0.21}$ | $14.68_{0.09}$ |
| | LA | $3.78_{0.60}$ | $7.63_{1.71}$ | $8.92_{4.16}$ | $11.97_{5.97}$ | $11.48_{1.66}$ |
| | BLoB(Mean) | $7.54_{0.57}$ | $4.89_{0.32}$ | $11.26_{1.13}$ | $6.83_{0.90}$ | $12.74_{0.88}$ |
| | BLoB(N=10) | $2.76_{0.41}$ | $\mathbf{3.35_{0.50}}$ | $6.81_{1.43}$ | $3.84_{1.00}$ | $11.96_{2.57}$ |
| | UQ4CT | $\mathbf{1.79_{0.43}}$ | $3.97_{0.78}$ | $\mathbf{4.43_{0.82}}$ | $\mathbf{3.34_{1.60}}$ | $\mathbf{9.36_{2.77}}$ |

modifying them. Checkpoint ensemble (Chen et al., 2017) utilizes predictions from multiple LoRA checkpoints saved during a single fine-tuning process to calibrate uncertainty. Deep ensemble (Lakshminarayanan et al., 2017; Wang et al., 2023; Zhai et al., 2023) combines the predictions from multiple LoRA adapters for better uncertainty calibration. Laplace-LoRA (Yang et al., 2024a) applies Bayesian inference via Laplace approximation to the LoRA parameters after fine-tuning, resulting in improved calibration and uncertainty estimates. Bayesian Low-Rank Adaptation by Backpropagation (BLoB) (Wang et al., 2024) extends the LA method by jointly optimizing the mean and covariance of LoRA parameters via backpropagation throughout fine-tuning.

Prompt-perturbation and resampling-based approaches have also been explored for uncertainty quantification in LLMs. These methods estimate uncertainty by measuring prediction variability under different prompt formulations or sampled input variants, without altering model parameters (Farquhar et al., 2024; Hou et al., 2023; Gao et al., 2024). This line of work leverages the inherent sensitivity of LLMs to input perturbations as a means to assess model confidence, providing a complementary perspective to parameter-based methods. Ye et al. (2024) benchmark LLMs using conformal prediction, which quantifies uncertainty by constructing prediction sets with guaranteed coverage, where set size directly reflects model uncertainty.

## 5 EXPERIMENTS

**Datasets.** We evaluate on five multiple-choice QA benchmarks: OpenBookQA (OBQA) (Mihaylov et al., 2018), ARC-Easy (ARC-E) and ARC-Challenge (ARC-C) (Clark et al., 2018), BOOLQ (Clark et al., 2019), and ClimateQA—a domain-specific climate science benchmark. To assess robustness under distribution shift, we ensemble the domain-specific MMLU subtasks (Hendrycks et al., 2020) into 4 benchmarks focusing on different professionalities: Computer Science (CS), Engineering (Eng), Law and Health. Details for the ensemble are provided in Appendix A.7. Models are fine-tuned on the public training split and evaluated on the test split for each benchmark.

**Experiment Setup.** We implement UQ4CT with PyTorch (Paszke et al., 2019), extending the MixLoRA repository in (Li et al., 2024). We use the Llama-3.1-8B (Touvron et al., 2023c) as our base model. In particular, we apply MixLoRA to query, key, value and output layers, together with the feed-forward networks in LLaMA-3.1-8B (gate layer, down layer and up layer). Details are provided in Appendix A.5.

**Baselines.** We compare UQ4CT with state-of-the-art uncertainty estimation methods along with naive fine-tuning applied to the LoRA adapters of LLMs, including **LoRA** (Hu et al., 2021a), **Monte**

Table 2: Performance comparison of different methods fine-tuned on the OBQA dataset with Llama3.1-8B across two smaller distribution shift (DS) tasks and four larger distribution shift tasks. UQ4CT shows substantial ECE improvements while maintaining high accuracy.

| Metrics | Methods | ID | Smaller DS | | Larger DS | | | |
|---|---|---|---|---|---|---|---|---|
| | | OBQA | ARC-C | ARC-E | CS | Eng | Law | Health |
| ACC ↑ | LoRA | $88.0_{1.22}$ | $77.8_{0.16}$ | $86.7_{0.77}$ | $\mathbf{55.8_{0.52}}$ | $54.3_{3.30}$ | $44.9_{0.23}$ | $58.8_{0.21}$ |
| | MC Drop | $87.1_{1.18}$ | $77.1_{2.01}$ | $86.9_{2.42}$ | $54.4_{1.58}$ | $54.1_{1.82}$ | $45.0_{0.76}$ | $58.3_{1.46}$ |
| | Ensemble | $86.5_{0.42}$ | $78.2_{0.90}$ | $85.4_{0.47}$ | $53.8_{1.02}$ | $52.4_{0.56}$ | $45.0_{0.20}$ | $60.6_{0.57}$ |
| | MixLoRA | $88.3_{0.50}$ | $78.1_{0.45}$ | $86.7_{0.35}$ | $53.1_{1.14}$ | $54.7_{2.28}$ | $45.0_{1.46}$ | $60.9_{1.04}$ |
| | LA | $86.0_{6.01}$ | $78.7_{0.55}$ | $86.4_{0.76}$ | $54.7_{1.82}$ | $53.6_{2.77}$ | $44.9_{1.03}$ | $59.7_{0.94}$ |
| | BLoB(Mean) | $87.6_{1.04}$ | $79.5_{1.10}$ | $86.6_{0.65}$ | $51.2_{0.99}$ | $48.6_{1.44}$ | $39.9_{7.85}$ | $57.0_{3.51}$ |
| | BLoB(N=10) | $87.1_{0.88}$ | $\mathbf{79.8_{1.06}}$ | $87.2_{0.79}$ | $52.8_{1.28}$ | $51.9_{3.13}$ | $43.8_{4.60}$ | $58.5_{5.33}$ |
| | UQ4CT | $\mathbf{88.4_{0.35}}$ | $79.0_{0.56}$ | $\mathbf{87.8_{0.47}}$ | $53.3_{0.61}$ | $\mathbf{61.1_{3.20}}$ | $\mathbf{45.4_{0.50}}$ | $\mathbf{61.1_{1.48}}$ |
| ECE ↓ | LoRA | $7.30_{0.43}$ | $14.8_{0.62}$ | $9.6_{0.69}$ | $21.0_{3.05}$ | $24.1_{3.63}$ | $29.3_{1.98}$ | $24.0_{1.81}$ |
| | MC Drop | $7.24_{0.39}$ | $13.4_{2.15}$ | $10.2_{1.89}$ | $20.8_{3.26}$ | $24.1_{0.77}$ | $29.1_{0.64}$ | $21.6_{3.92}$ |
| | Ensemble | $8.63_{0.38}$ | $15.4_{0.46}$ | $10.7_{0.56}$ | $14.0_{3.18}$ | $17.4_{1.98}$ | $19.9_{2.95}$ | $16.1_{2.07}$ |
| | MixLoRA | $6.58_{0.21}$ | $14.5_{0.55}$ | $9.9_{0.20}$ | $17.1_{3.06}$ | $17.8_{2.80}$ | $21.6_{4.07}$ | $18.0_{2.78}$ |
| | LA | $11.97_{5.97}$ | $7.2_{0.5}$ | $6.4_{0.42}$ | $13.7_{2.14}$ | $15.5_{2.0}$ | $19.0_{0.54}$ | $15.7_{2.30}$ |
| | BLoB(Mean) | $6.83_{0.90}$ | $11.37_{1.94}$ | $6.6_{1.65}$ | $17.2_{2.72}$ | $18.5_{2.82}$ | $22.6_{1.26}$ | $16.9_{3.02}$ |
| | BLoB(N=10) | $3.84_{1.00}$ | $5.8_{0.96}$ | $\mathbf{3.0_{0.87}}$ | $11.5_{2.76}$ | $14.9_{1.93}$ | $19.7_{3.21}$ | $14.5_{3.38}$ |
| | UQ4CT | $\mathbf{3.34_{1.60}}$ | $\mathbf{3.6_{1.44}}$ | $3.6_{1.32}$ | $\mathbf{10.8_{3.73}}$ | $\mathbf{13.2_{1.86}}$ | $18.1_{4.40}$ | $\mathbf{13.2_{4.06}}$ |

**Carlo (MC) Dropout** (Gal & Ghahramani, 2016), **Deep Ensemble** (Lakshminarayanan et al., 2017), **Laplace-LoRA (LA)** (Yang et al., 2024a), **Bayesian Low-Rank Adaptation by Backpropagation (BLoB)** (Wang et al., 2024) and **MixLoRA** (Li et al., 2024). Note that for **BLoB(N=10)**, the method performs 10 forward passes with differently sampled LoRA parameters for each question, which is a unfair computational budget advantage compared against UQ4CT with only 1 forward pass.

**Evaluation.** We evaluate prediction accuracy on the validation set across all five tasks. For uncertainty calibration, we use the expected calibration error (ECE; Guo et al. (2017); more details in A.6) to measure the alignment between predicted probabilities and actual outcomes.

To assess robustness under distribution shifts, we fine-tune models on the OBQA dataset and evaluate them following Yang et al. (2024a). We use ARC-C and ARC-E to represent smaller distribution shifts, as these datasets focus on general science reasoning similar to OBQA but are more challenging and diverse. For larger shifts, we utilize the four aforementioned domain-specific MMLU subtasks, which span a wide range of expertise from elementary to professional levels. This domain specificity represents a greater distribution shift from the general common sense focus of OBQA.

The in-distribution scenario tests model alignment on the target task, while the distribution shift scenario assesses generalizability to novel domains. Together, they provide a comprehensive evaluation for real-world applications, ensuring strong performance on the primary task and resilience to out-of-distribution inputs.

## 5.1 IN-DISTRIBUTION PERFORMANCE

As shown in Table 1, UQ4CT achieves notable gains in uncertainty calibration across diverse tasks, while maintaining competitive accuracy (ACC) relative to baseline approaches. For instance, on OBQA and ClimateQA tasks, UQ4CT attains accuracy rates of 88.4% and 79.9%, demonstrating that improved uncertainty quantification does not come at the expense of predictive performance.

The most significant improvements are seen in reduced Expected Calibration Error (ECE). UQ4CT consistently lowers ECE by over 25% on average across benchmarks, and unlike other approaches, it continues to perform well even on challenging datasets such as ARC-C, achieving an ECE of 4.4.

To further validate our method, we include results from fine-tuning both LLaMA-3.1-8B (main text) and Mistral-7B (Appendix A.3). Across both models, UQ4CT delivers substantial and consistent improvements in uncertainty calibration. These results underscore the practical value of UQ4CT, particularly in scenarios where reliable uncertainty estimates are crucial, such as safety-critical applications. A key advantage of UQ4CT is that it incorporates uncertainty calibration directly during

Table 3: Performance of UQ4CT with varying $\beta$ values on the OBQA dataset. Prediction accuracy and uncertainty calibration improve with increasing $\beta$, highlighting the effectiveness of $\mathcal{L}_{\text{cal}}$.

| $\beta$ | 0 | 0.2 | 0.5 | 0.8 | 1 | 1.2 | 1.5 | 1.8 | 2 |
|---|---|---|---|---|---|---|---|---|---|
| **ACC** ↑ | $87.0_{2.85}$ | $87.1_{0.58}$ | $87.1_{0.29}$ | $87.3_{0.38}$ | $\mathbf{88.4_{0.35}}$ | $88.3_{0.57}$ | $87.5_{0.88}$ | $87.2_{0.79}$ | $87.3_{0.89}$ |
| **ECE** ↓ | $12.7_{1.92}$ | $7.35_{0.75}$ | $7.69_{0.89}$ | $5.82_{1.12}$ | $\mathbf{3.34_{1.60}}$ | $6.31_{2.58}$ | $9.03_{0.82}$ | $7.96_{1.38}$ | $6.52_{1.79}$ |

fine-tuning, incurring minimal computational overhead compared to other uncertainty quantification (UQ) methods, which often require costly repetitive sampling or post-hoc adjustments.

## 5.2 PERFORMANCE UNDER DISTRIBUTION SHIFT

Due to the sparse nature of the fine-tuning data, real world deployment of LLMs often requires the model to be robust to out-of-distribution knowledge (Ouyang et al., 2022; Touvron et al., 2023b;c). Therefore, we evaluate the performance of UQ4CT along with other baseline models fine-tuned on the OBQA dataset under smaller and larger distribution shift scenarios.

Table 2 presents the distribution shift evaluations. UQ4CT achieves substantial ECE improvements while maintaining high accuracy across both smaller and larger distribution shifts. For smaller shifts, UQ4CT's ECE remains comparable to the in-distribution scenario. Under larger shifts, UQ4CT attains the lowest ECE among all baselines and delivers competitive accuracy on all domain-specific tasks. These results demonstrate that aligning uncertainty at the functional level with predictive correctness improves generalizability and mitigates overconfidence in fine-tuned models.

## 5.3 ABLATION STUDIES

We conduct ablation studies to investigate the effectiveness of our designed calibration loss, $\mathcal{L}_{cal}$. We first perform a sensitivity test, in which we explore the impact of $\mathcal{L}_{cal}$ on the overall performance. Then we evaluate the incremental weighting performance of the calibration term , which investigates the effectiveness of $\mathcal{L}_{cal}$ at the early stage of fine-tuning. We also conduct an ablation study on the impact of active LoRA experts in Appendix A.4. We compare our method with prompt perturbation based method in AppendixA.8.

**Sensitivity Test on Calibration Term.** To further understand the effectiveness of the calibration loss, we perform a sensitivity test of $\beta$ in Equation 14. This evaluates how our proposed calibration of parameter mixtures affect the overall model prediction and uncertainty quantification capabilities. We evaluate $\beta$ values ranging from 0 to 2, where $\beta = 0$ resembles the original MixLoRA method.

Results in Table 3 demonstrate the effectiveness of the calibration loss. When $\beta = 0$, the model is optimized without calibration on parameter mixtures, resulting in high ECE value. Even with small $\beta = 0.2$ or $\beta = 0.5$, the ECE scores drastically improved compared to no calibration setting. Finally, when $\beta = 1$, the calibration term effectively optimizes the conditional parameter mixtures to generate outputs that fit data distribution well, resulting in lower ECE scores and higher accuracies.

**Incremental Weighting on Calibration Term.** Due to the random initialization of LoRA experts, the predictions during early fine-tuning stage are likely to be incorrect as the model has little knowledge on the functional relationships regarding the data. Thus, it is intuitive to incrementally increase the weight parameter $\beta$ over the calibration term $\mathcal{L}_{\text{cal}}$ in the training loss for the LoRA experts to learn before calibration. We conduct this study by incrementally increase $\beta$ from 0 to 1 within 50 gradient steps during the early stage of fine-tuning:

$$\beta = \min\left\{1, \frac{\text{current\_grad\_step}}{50}\right\}.$$ (15)

We choose 50 gradient steps from our observation that training loss generally stabilizes after 50 gradient steps, indicating the LoRA experts have learned some functional relationships from data.

As shown in Table 4, the incremental loss has significantly worse ECE performance across all tasks. This demonstrates the advantage of uncertainty calibration even in the early stage. In the beginning, the lack of functional relationships on the training data in LoRA experts lead to high epistemic

Table 4: Performance comparison of UQ4CT with and without incremental weighting. Incremental weighting has worse ECE performance while maintains similar accuracy.

| Metrics | Methods | BoolQ | ARC-E | ARC-C | OBQA | ClimateQA |
|---------|---------|-------|-------|-------|------|-----------|
| ACC $\uparrow$ | UQ4CT | $89.17_{1.33}$ | $88.66_{0.20}$ | $79.60_{1.21}$ | $88.40_{0.35}$ | $79.97_{0.85}$ |
| | UQ4CT_Incremental | $87.33_{0.24}$ | $87.15_{0.95}$ | $80.84_{1.03}$ | $88.53_{0.48}$ | $75.87_{2.89}$ |
| ECE $\downarrow$ | UQ4CT | $\mathbf{1.79_{0.43}}$ | $\mathbf{3.97_{0.78}}$ | $\mathbf{4.43_{0.82}}$ | $\mathbf{3.34_{1.60}}$ | $\mathbf{9.36_{2.77}}$ |
| | UQ4CT_Incremental | $3.10_{0.18}$ | $6.55_{1.42}$ | $10.02_{1.95}$ | $6.87_{1.68}$ | $14.16_{0.91}$ |

uncertainty. Thus, UQ4CT encourages exploration over all LoRA experts while UQ4CT_Incremental lacks it due to the small weighting in the beginning.

## 6 DISCUSSION & CONCLUSION

In this work, we propose Functional-Level Uncertainty Quantification for Calibrated Fine-Tuning (UQ4CT), which addresses the overconfidence issues commonly encountered during fine-tuning of large language models. We present a functional perspective on quantifying uncertainty in LLMs and utilize it for uncertainty-calibrated fine-tuning. By incorporating functional-level uncertainty quantification with a mixture-of-experts framework, our proposed uncertainty-calibrated training loss effectively addresses the challenge of overconfidence in fine-tuned LLMs by significantly improving uncertainty calibration while maintaining high accuracy. Our evaluations demonstrate that UQ4CT reduces the Expected Calibration Error by more than 25% without compromising accuracy across a variety of downstream tasks, including common-sense and domain-specific reasoning, under in-distribution and out-of-distribution scenarios.

The limitation of UQ4CT lies in its dependency on predictive correctness. For general language modeling tasks such as chat completion, there lacks a clear metric on response correctness. This limits the application of UQ4CT as naively token matching is a poor indicator of semantic correctness due to the ambiguous nature of language. For future work, we are exploring ways to adapt UQ4CT to open-ended problems that lack a definitive optimization objective.

## 7 REPRODUCIBILITY STATEMENT

We will make all code, simulators, and benchmark datasets publicly available to ensure reproducibility. A code repository is included in the supplementary materials and will be released upon paper acceptance. Detailed implementation instructions are provided in the repository's README file.

## 8 ETHICS STATEMENT

Our work aims to advance the trustworthiness of large language models, which we foresee positive impacts in the applicable fields, such as medical advising and general reasoning. Since we are using public datasets, we do not foresee any ethic problems.

## 9 LLM USAGE

Large language models were used exclusively for refining the writing style. They were not employed for generating content or shaping ideas.

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

# A APPENDIX

## A.1 THEORETICAL DERIVATION OF THE METHOD

In this section, we provide complete theoretical statements and proofs that are used in Sec. 3 to derive our method.

**Fact A.1** (Model Perturbation Structure, Restatement of Fact 3.1)**.** *Assume that in the residual connection architecture in each layer:* $g_{k^\ell}^\ell(\mathbf{h}^\ell) = \mathbf{h}^\ell + E_{k^\ell}^\ell(f_{trans}^\ell(\mathbf{h}^\ell))$, *the Lipschitzness of the residual term* $E_{k^\ell}^\ell(f_{trans}^\ell(\mathbf{h}^\ell))$ *is much smaller as compared to that of* $\mathbf{h}^\ell$ *itself:* $\left\| E_{k^\ell}^\ell(f_{trans}^\ell(\hat{\mathbf{h}}^\ell)) - E_{k^\ell}^\ell(f_{trans}^\ell(\mathbf{h}^\ell)) \right\| = o\left( \|\hat{\mathbf{h}}^\ell - \mathbf{h}^\ell\| \right)$. *Under this regularity assumption, perturbations* $\Delta f(x)$ *to the model* $f(x)$ *approximately decomposes as:*

$$\Delta f(x) \approx \sum_{\ell=1}^{L} \sum_{k^\ell=1}^{K} \Delta\alpha_{k^\ell}^\ell(\mathbf{h}^\ell) \cdot g_{k^\ell}^\ell(\mathbf{h}^\ell).$$

*Proof of Fact 3.1 and A.1.* In each layer, we can decompose the perturbation to the output as follows:

$$\Delta\mathbf{h}^{\ell+1} = \sum_{k^\ell=1}^{K} \left[ \Delta\alpha_{k^\ell}^\ell(\mathbf{h}^\ell) \cdot g_{k^\ell}^\ell(\mathbf{h}^\ell) + \alpha_{k^\ell}^\ell(\mathbf{h}^\ell) \cdot \Delta g_{k^\ell}^\ell(\mathbf{h}^\ell) \right], \quad \forall\ell = 1, \ldots, L,$$

where the input $\mathbf{h}^1 = x$ and the output $f(x) = \mathbf{h}^{L+1}$.

Due to the residual connection architecture and our assumption on the regularity of the residual term, we have:

$$\Delta g_{k^\ell}^\ell(\mathbf{h}^\ell) = \Delta\mathbf{h}^\ell + \Delta E_{k^\ell}^\ell(f_{trans}^\ell(\mathbf{h}^\ell)) = \Delta\mathbf{h}^\ell + o\left(\Delta\mathbf{h}^\ell\right).$$

Hence,

$$\Delta\mathbf{h}^{\ell+1} \approx \sum_{k^\ell=1}^{K} \left( \Delta\alpha_{k^\ell}^\ell(\mathbf{h}^\ell) \cdot g_{k^\ell}^\ell(\mathbf{h}^\ell) + \alpha_{k^\ell}^\ell(\mathbf{h}^\ell) \cdot \Delta\mathbf{h}^\ell \right) = \sum_{k^\ell=1}^{K} \Delta\alpha_{k^\ell}^\ell(\mathbf{h}^\ell) \cdot g_{k^\ell}^\ell(\mathbf{h}^\ell) + \Delta\mathbf{h}^\ell.$$

Expanding this recursion, the output perturbation can be approximated as:

$$\Delta f(x) = \Delta\mathbf{h}^{L+1} \approx \sum_{\ell=1}^{L} \sum_{k^\ell=1}^{K} \Delta\alpha_{k^\ell}^\ell(\mathbf{h}^\ell) \cdot g_{k^\ell}^\ell(\mathbf{h}^\ell).$$

$\square$

**Proposition A.2** (Calibration Loss, Restatement of Proposition 3.2)**.** *Let the calibration risk be defined as the expectation of the calibration loss over the data distribution:*

$$\bar{\mathcal{L}}_{\text{cal}} = \mathbb{E}_{(x,y^*)\sim\mathcal{D}} \left( \mathbb{1}\{\text{MixLoRA}(x) = y^*\} - \text{FLC}(x) \right)^2.$$

*If this calibration risk is optimized over the data distribution* $\mathcal{D}$, *then the optimal solution is* $\text{FLC}(x) = \mathbb{P}(\text{MixLoRA}(x) = y^*(x))$; *that is, the optimally trained FLC corresponds to the probability that the model's prediction is correct.*

*Proof of Proposition 3.2 and A.2.* Expanding the calibration risk, we have:

$$\begin{aligned}
\bar{\mathcal{L}}_{\text{cal}} &= \mathbb{E}_{(x,y^*)\sim\mathcal{D}} \left( \mathbb{1}\{\text{MixLoRA}(x) = y^*\} - \text{FLC}(x) \right)^2 \\
&= \mathbb{E}_{(x,y^*)\sim\mathcal{D}} \left[ \mathbb{1}\{\text{MixLoRA}(x) = y^*\} \right]^2 \\
&\quad - 2\,\mathbb{E}_x\mathbb{E}_{y^*|x} \left[ \mathbb{1}\{\text{MixLoRA}(x) = y^*\} \cdot \text{FLC}(x) \right] + \mathbb{E}_x \left[ \text{FLC}(x)^2 \right] \\
&= C + \mathbb{E}_x \left[ \text{FLC}(x)^2 - 2\,\mathbb{P}(\text{MixLoRA}(x) = y^*(x)) \cdot \text{FLC}(x) \right],
\end{aligned}$$

where $C$ is a constant independent of $\text{FLC}(x)$. Thus, the calibration risk is minimized when $\text{FLC}(x) = \mathbb{P}(\text{MixLoRA}(x) = y^*(x))$.

$\square$

Table 5: Performance comparison of different methods fine-tuned with Mistral-7B across 4 common sense reasoning tasks and a domain-specific task. UQ4CT shows significant ECE improvements while maintaining high accuracy.

| Metrics | Methods | BoolQ | ARC-E | ARC-C | OBQA | ClimateQA |
|---------|---------|-------|-------|-------|------|-----------|
| ACC ↑ | LoRA | $70.3_{0.62}$ | $84.8_{0.47}$ | $70.2_{0.84}$ | $82.8_{0.62}$ | $72.5_{1.6}$ |
| | MC Drop | $69.6_{1.07}$ | $84.6_{0.91}$ | $69.6_{0.76}$ | $82.6_{0.71}$ | $72.5_{1.6}$ |
| | Ensemble | $71.8_{1.29}$ | $84.2_{0.66}$ | $71.0_{1.41}$ | $82.5_{0.6}$ | $72.9_{2.88}$ |
| | LA | $70.7_{1.82}$ | $82.4_{2.05}$ | $68.5_{3.31}$ | $82.5_{0.77}$ | $71.6_{1.56}$ |
| | MixLoRA | $73.1_{0.38}$ | $85.5_{1.27}$ | $71.2_{1.75}$ | $83.3_{1.14}$ | $72.0_{1.69}$ |
| | UQ4CT | $\mathbf{73.6_{0.28}}$ | $\mathbf{85.9_{0.82}}$ | $\mathbf{74.4_{0.82}}$ | $\mathbf{83.7_{1.22}}$ | $\mathbf{73.2_{1.29}}$ |
| ECE ↓ | LoRA | $10.17_{0.24}$ | $9.46_{1.62}$ | $18.42_{1.91}$ | $13.3_{0.25}$ | $13.72_{2.62}$ |
| | MC Drop | $10.62_{0.51}$ | $8.91_{1.35}$ | $18.38_{1.66}$ | $13.3_{0.31}$ | $13.72_{2.61}$ |
| | Ensemble | $8.72_{1.13}$ | $8.72_{1.49}$ | $17.0_{0.97}$ | $9.14_{2.82}$ | $12.86_{1.78}$ |
| | LA | $5.33_{2.16}$ | $20.3_{5.7}$ | $21.27_{4.15}$ | $\mathbf{6.41_{3.22}}$ | $14.64_{2.21}$ |
| | MixLoRA | $8.81_{1.03}$ | $8.16_{0.99}$ | $15.51_{3.86}$ | $10.53_{1.73}$ | $14.05_{3.09}$ |
| | UQ4CT | $\mathbf{3.07_{0.83}}$ | $\mathbf{5.7_{0.69}}$ | $\mathbf{7.04_{0.58}}$ | $\underline{7.92_{1.14}}$ | $\mathbf{11.4_{1.14}}$ |

## A.2 LOAD BALANCING LOSS

We follow the load balancing loss in (Li et al., 2024). Given $N$ experts indexed by $i = 1$ to $N$ and a batch $B$ with $T$ tokens, the auxiliary loss is computed as:

$$\mathcal{L}_{aux} = a \cdot N \cdot \sum_{i=1}^{N} \mathcal{F}_i \cdot \mathcal{P}_i, \tag{16}$$

where

$$\mathcal{F}_i = \frac{1}{T} \sum_{x \in B} \mathbb{1}\{argmax_k \mathcal{R}(x)_k = i\}, \mathcal{P}_i = \frac{1}{T} \sum_{x \in B} \mathcal{R}(x)_i. \tag{17}$$

Here, $\mathcal{R}(\cdot)$ is the top-k router, $\mathcal{F}_i$ is the fraction of tokens dispatched to expert $i$ and $\mathcal{P}_i$ is the fraction of the router probability allocated for expert $i$. The final loss is multiplied by the expert count $N$ to keep the loss constant as the number of experts varies, and the constant term $a$ is set to $10^{-2}$ as a multiplicative coefficient, which is large enough to ensure load balancing while remaining small enough not to overwhelm the primary objective.

## A.3 EXPERIMENTAL RESULTS WITH MISTRAL-7B

In this section, we present the results using Mistral-7B (Jiang et al., 2023), a different decoder-based LLM backbone. Table 5 shows the results of fine-tuning Mistral-7B on 4 common-sense reasoning tasks and one domain-specific climate question-answering task.

For each of the tasks, UQ4CT effectively calibrates the parameter mixtures, leading to the best ECE performance in 4 out of 5 tasks. This indicates the robustness of UQ4CT across different LLMs.

## A.4 DECIDING NUMBER OF ACTIVE EXPERTS

One important aspect of the LoRA MoE architecture is how many experts to activate. Here, we investigate the performance impact of different number of active LoRA experts. We evaluate the model performance with 1 to 5 active experts with 8 in total.

As shown in Table 6, 2 active experts give the optimal performance in terms of accuracy and ECE scores. One expert alone cannot capture complicated functional relationships, while more than 2 experts could potentially introduce redundant functional bases to the model, which deviates the output distribution more from data distribution, thus worsening predictive and calibration performance. Additionally, more active experts lead to a more flattened distribution across experts, which hardens the alignment of parameter mixtures during fine-tuning.

Table 6: Performance comparison of UQ4CT with varying number of experts on OBQA dataset. Top-2 expert selection strategy grants best accuracy and calibration.

| Top-K | ACC $\uparrow$ | ECE $\downarrow$ |
|-------|------|------|
| Top-1 | $86.8_{0.59}$ | $7.54_{1.89}$ |
| Top-2 | $\mathbf{88.4_{0.35}}$ | $\mathbf{3.34_{1.60}}$ |
| Top-3 | $87.0_{0.77}$ | $5.68_{0.78}$ |
| Top-4 | $87.4_{0.51}$ | $7.52_{0.44}$ |
| Top-5 | $87.1_{0.48}$ | $6.16_{0.58}$ |

## A.5 TRAINING DETAILS

We train our model with total of $8$ LoRA experts, and select $2$ experts with the highest probability. For each expert, we use $rank = 16$ and $alpha = 32$. We use batch size of $16$ to train our model. For climate task, we set the learning rate to $5e - 4$ and dropout rate to $0.1$ to incorporate the small dataset size. For other tasks, we use $2e - 4$ as our learning rate with dropout $0.05$. We use AdamW as our optimizer and a cutoff length of $512$ for prompts during training. Our model is trained on A100 GPU, with 20GB GPU memory consumption per task. Training time is from $25$ to $50$ minutes depending on the task.

The experimental setup for single LoRA based models is similar with LoRA ranks set to $80$ to accommodate the MoE model size. For the ensemble baseline, we use an ensemble size of $8$ with $rank = 16$. For Laplace-LoRA, we follow the Laplace hyperparameters in this Github Repository.

## A.6 EXPECTED CALIBRATION ERROR

Expected calibration error (ECE) is a commonly used metric to asses uncertainty quantification performance. ECE measures the alignment between prediction accuracy and model confidence through regrouping the predicted probabilities into $m$ bins. This method then computes the weighted average of the difference between average accuracy and confidence in each bin:

$$\text{ECE} = \sum_{m=1}^{M} \frac{|B_m|}{N} |\text{acc}(B_m) - \text{conf}(B_m)|, \tag{18}$$

where $|B_m|$ is the number of evaluated datapoints in bin $m$, acc and conf is calculated as following:

$$\text{acc}(B_m) = \frac{1}{|B_m|} \sum_{i \in B_m} \mathbf{1}(\hat{y}_i = y_i), \tag{19}$$

$$\text{conf}(B_m) = \frac{1}{|B_m|} \sum_{i \in B_m} P(\hat{y}_i). \tag{20}$$

In this paper, we use an ECE bin size of $15$, following the experiment setup in Laplace-LoRA (Yang et al., 2024a).

## A.7 MMLU DISTRIBUTION SHIFT DATASET COMPOSITION

- **Computer Science (CS)**:
    - College Computer Science
    - Computer Security
    - High School Computer Science
    - Machine Learning
- **Engineering (Eng)**:
    - Electrical Engineering
- **Law**:
    - International Law

– Jurisprudence
– Professional Law

- **Health**:
  – Anatomy
  – Clinical Knowledge
  – College Medicine
  – Human Aging
  – Nutrition
  – Professional Medicine
  – Virology

## A.8 PROMPT PERTURBATION COMPARISON

Here, we compare our method with SPUQ (Gao et al., 2024), which perturbs the prompt, aggregates predictions and confidences to measure uncertainty. We test SPUQ and UQ4CT with LLama3.1-8b as the base model. As shown in Table 7, SPUQ's large ECE values suggest that simply aggregating predictions from perturbed prompts does not adequately calibrate model confidence, highlighting the limitations of prompt perturbation as an uncertainty quantification strategy for LLMs, especially for smaller models.

Table 7: Performance comparison of UQ4CT and SPUQ across five tasks. UQ4CT achieves higher accuracy and substantially lower ECE than SPUQ.

| Metrics | Methods | BoolQ | ARC-E | ARC-C | OBQA | ClimateQA |
|---------|---------|-------|-------|-------|------|-----------|
| ACC $\uparrow$ | SPUQ | 73.79 | 86.67 | 73.94 | 74.00 | 67.80 |
| | UQ4CT | $89.17_{1.33}$ | $88.66_{0.20}$ | $79.60_{1.21}$ | $88.40_{0.35}$ | $79.97_{0.85}$ |
| ECE $\downarrow$ | SPUQ | 7.70 | 10.81 | 7.40 | 8.86 | 14.68 |
| | UQ4CT | $\mathbf{1.79_{0.43}}$ | $\mathbf{3.97_{0.78}}$ | $\mathbf{4.43_{0.82}}$ | $\mathbf{3.34_{1.60}}$ | $\mathbf{9.36_{2.77}}$ |

