# OpenReview forum: "Functional-level Uncertainty Quantification for Calibrated Fine-tuning on LLMs"
_ICLR.cc/2026/Conference — Submitted to ICLR 2026_

### Official Review · Reviewer_yiFV · 2025-10-25

**Soundness:** 2
**Presentation:** 3
**Contribution:** 3
**Rating:** 4
**Confidence:** 4

**Summary:**

The paper presents a new method for quantifying uncertainty in large language models. The proposed method focuses on quantifying functional uncertainty using a mixture of LoRA experts to all parameter matrices during finetuning. The model is deemed to be more confident for a particular generation given an input if it assigns high probability to a single expert. The overall functional uncertainty score computed across all model trainable components is incorporated as an auxiliary loss function during fine-tuning. Overall experimental results show that the proposed approach offers better uncertainty estimation and high performance retention using a Llama-3.1 8B model across several standard datasets.

**Strengths:**

- Really nice and creative approach for efficiently calibrating LLMs during fine-tuning.

**Weaknesses:**

- While I really liked the method, the experiments are limited. Only a single model size from a single family is tested. Furthermore, fine-tuning on the datasets you used in Table 2 or OBQA is not realistic (see my suggestions below). The current experimental setting does not convincingly show if the proposed method generalizes.

**Questions:**

1. You should report results using at least one more model family (e.g., Qwen3, OLMO 2 or similar) and model size (e.g. 3B or larger than 8B if you have enough compute).

2. You should use standard instruction tuning data (Alpaca or similar) to SFT your models instead of your current setting.

---

> ### Author Response · Authors · 2025-11-24
>
> > While I really liked the method, the experiments are limited. Only a single model size from a single family is tested. Furthermore, fine-tuning on the datasets you used in Table 2 or OBQA is not realistic (see my suggestions below). The current experimental setting does not convincingly show if the proposed method generalizes.
>
> > You should report results using at least one more model family (e.g., Qwen3, OLMO 2 or similar) and model size (e.g. 3B or larger than 8B if you have enough compute).
>
> We understand your concern, as we did include results for an additional model family, Mistral-7B, in appendix A.3 to demonstrate the generality of our approach.
>
> > You should use standard instruction tuning data (Alpaca or similar) to SFT your models instead of your current setting.
>
> We appreciate the suggestion. In our setting, using the base model is sufficient and aligns with prior work on uncertainty for PEFT methods, as both BLoB and Laplace-LoRA also evaluate on base models without additional instruction tuning. Since our goal is to analyze functional-level confidence after fine-tuning on specific tasks rather than improve instruction-following ability in general, we believe adding Alpaca-style SFT is unnecessary and would confound the goal of our method and the comparison with the existing works.

---

### Official Review · Reviewer_d28n · 2025-10-28

**Soundness:** 2
**Presentation:** 1
**Contribution:** 2
**Rating:** 4
**Confidence:** 3

**Summary:**

This paper proposes a fine-tuning approach for calibrating MixLoRA-adapted LLMs. The authors define functional uncertainty (FLU) as the predictive difference between the adapted model and its vanilla LLM counterpart. The paper then shows that this FLU can be linearly expressed by the MoE weights in the MixLoRA architecture. To calibrate this FLU score, the paper introduces a Brier-score-like calibration loss as the auxiliary loss.

**Strengths:**

1. The analysis presented in Fact 3.1 is insightful. The resulting equation, which linearly approximates the perturbation using $\Delta \alpha(h)$ and $g(h)$ (omitting super- and subscripts for simplicity), is surprising.
2. The experiments are basically comprehensive, including both in-distribution and out-of-distribution evaluations, and the reported calibration performance appears strong.
3. The methodology does not require modifications to the model's architecture, making it agnostic to the underlying vanilla LLM.

**Weaknesses:**

1. The derivation of uncertainty in Fact 3.1, defined as the difference between the fine-tuned (MixLoRA-adapted) LLM and the vanilla LLM, is counter-intuitive. The authors should clarify which model is the subject of the uncertainty analysis. If, as understood, it is the fine-tuned LLM, a more logical definition of uncertainty would seem to involve a comparison between the MixLoRA LLM and a perturbed version of itself, rather than a comparison against the vanilla LLM.

2. In Fact 3.1, $\Delta f(x)$ is expressed using both $\Delta \alpha(h)$ and $g(h)$. However, $g(h)$ is omitted from the subsequent definition of FLU. The authors should justify this omission.

3. The definition of FLU in Eq. (12), the calibration loss in Eq. (13), and Proposition (3.2) are highly counter-intuitive. Eq. (12) appears to model the confidence that "Expert $i$ should be involved in the computation," and the calibration loss (Eq. 13) reinforces this interpretation. This, however, represents confidence, not uncertainty. This contradiction is evident from Proposition (3.2): a perfect (100% accurate) fine-tuned LLM would have optimal MoE selection, implying zero uncertainty. In contrast, the proposed formulation would result in an FLU of 1.

4. The computational budget comparison with BLOB (Lines 399-402) is not straightforward. BLOB requires multiple forward passes, whereas the proposed method utilizes an MoE structure.
Given these fundamental architectural differences, a more rigorous and detailed complexity analysis is needed to make the comparison solid.

5. Figure 2 requires polishing, and its caption is misleading. Furthermore, when referenced in the text (e.g., Lines 273-274), there is a lack of proper guidance for the reader to interpret it.

6. The Introduction and Related Work sections could be strengthened by discussing other lines of LLM calibration, such as evidential methods [1], to provide a more complete background.

[1] Li, Yawei, et al. "Calibrating LLMs with Information-Theoretic Evidential Deep Learning." ICLR 2025.

**Questions:**

Please see the Weaknesses.

---

> ### Author Response · Authors · 2025-11-24
>
> We appreciate the insightful and constructive review. We address the weaknesses and questions as follows:
>
> > The derivation of uncertainty in Fact 3.1, defined as the difference between the fine-tuned (MixLoRA-adapted) LLM and the vanilla LLM, is counter-intuitive. The authors should clarify which model is the subject of the uncertainty analysis. If, as understood, it is the fine-tuned LLM, a more logical definition of uncertainty would seem to involve a comparison between the MixLoRA LLM and a perturbed version of itself, rather than a comparison against the vanilla LLM.
>
> The uncertainty in Fact 3.1 is defined relative to the base model because MixLoRA creates a family of perturbed functions: different prompts activate different expert weights, producing different low-rank updates on top of the same shared backbone. The variation induced by these prompt-dependent expert combinations is therefore naturally measured as deviation from the base model. In this sense, the “perturbation” is already generated within MixLoRA through its routing, and comparing against the base model captures exactly this functional shift; comparing a fixed MixLoRA instance to a perturbed version of itself would not reflect the source of variability that MixLoRA introduces.
>
> > In Fact 3.1,  ∆f(x) is expressed using both ∆α(h) and g(h). However, g(h)  is omitted from the subsequent definition of FLU. The authors should justify this omission.
>
> This omission is justified in Fact 3.1: in high-dimensional settings the basis functions g(h) are approximately orthogonal, so the functional perturbation Δf(x) is determined by the change in the mixing coefficients Δα(h). Since g(h) represents the fixed learned basis and the variability comes solely from α, we omit g(h) in the subsequent FLU definition. We have included further explanation in the revised version to clear this ambiguity.
>
> > The definition of FLU in Eq. (12), the calibration loss in Eq. (13), and Proposition (3.2) are highly counter-intuitive. Eq. (12) appears to model the confidence that "Expert  should be involved in the computation," and the calibration loss (Eq. 13) reinforces this interpretation. This, however, represents confidence, not uncertainty. This contradiction is evident from Proposition (3.2): a perfect (100% accurate) fine-tuned LLM would have optimal MoE selection, implying zero uncertainty. In contrast, the proposed formulation would result in an FLU of 1.
>
> Thank you for pointing this out. The issue arises from naming rather than formulation: FLU represents the model’s confidence in the selected functional perturbation, not uncertainty itself. We agree that this terminology may be misleading. In the revision, we have renamed FLU to FLC (Functional-Level Confidence) to better reflect its meaning and avoid the confusion.
>
> > The computational budget comparison with BLOB (Lines 399-402) is not straightforward. BLOB requires multiple forward passes, whereas the proposed method utilizes an MoE structure. Given these fundamental architectural differences, a more rigorous and detailed complexity analysis is needed to make the comparison solid.
>
> For BLoB (N=10), the method uses plain LoRA and requires 10 independent forward passes (cost = 10 × K, where K is the LoRA parameter size). In contrast, our MoE-based approach activates only 2 experts per input (top-2 routing), so the total computation per sample is 2 × K (assuming same LoRA parameter size), with negligible router overhead. Thus, our approach achieves comparable uncertainty quantification at approximately 1/5 the computational cost compared to BLoB(N=10), given the same expert size. We will clarify this distinction and provide a more explicit complexity comparison in the revision.
>
> > Figure 2 requires polishing, and its caption is misleading. Furthermore, when referenced in the text (e.g., Lines 273-274), there is a lack of proper guidance for the reader to interpret it.
>
> We believe the confusion in Figure 2 and its caption primarily stems from the same naming issue (FLU → FLC). Since the figure visualizes router probabilities, the updated terminology clarifies its interpretation. In the revision, we have corrected the naming, and improved the textual guidance accordingly.
>
> > The Introduction and Related Work sections could be strengthened by discussing other lines of LLM calibration, such as evidential methods [1], to provide a more complete background.
>
> We appreciate the suggestion and have added this in the updated version.

---

### Official Review · Reviewer_6Wcn · 2025-10-30

**Soundness:** 2
**Presentation:** 2
**Contribution:** 2
**Rating:** 4
**Confidence:** 3

**Summary:**

This paper proposes a new approach to uncertainty estimation and calibration in large language models during fine-tuning. Instead of modeling uncertainty at the parameter level or requiring multiple stochastic forward passes, the authors introduce functional-level uncertainty , which quantifies uncertainty directly from the Mixture-of-Experts routing probabilities in the model.

**Strengths:**

1. The paper tackles a timely and interesting topic.
2. The proposed functional-level uncertainty formulation, leveraging MoE routing probabilities, is conceptually novel and computationally efficient.

**Weaknesses:**

1. Limited application scope. The method only applies to Mixture-of-Experts (MoE) architectures and to discriminative tasks with ground-truth labels. It is not directly applicable to generative or open-ended tasks, which limits its broader relevance in LLM research.
2. Theoretical justification is oversimplified. A more rigorous or relaxed analysis would strengthen the paper.
3. Comparable performance to BLoB (N=10). Although the proposed method is more efficient (single-pass), its calibration and accuracy improvements over BLoB(N=10) are modest, suggesting a trade-off between novelty and practical gains.

**Questions:**

1. Unclear token-level aggregation. In Eq. (12), the aggregation of routing probabilities across tokens is not well defined — it should specify whether it averages over all tokens, the [CLS] token, or the last token.
2. Missing ablation results in main text. The ablation study on the calibration loss weight (Lines 454–458) should be moved from the appendix to the main body for better visibility, since it demonstrates the effectiveness of the calibration component.

---

> ### Author Response · Authors · 2025-11-24
>
> We appreciate the insightful and constructive review. We address the weaknesses and questions as follows:
>
> > Limited application scope. The method only applies to Mixture-of-Experts (MoE) architectures and to discriminative tasks with ground-truth labels. It is not directly applicable to generative or open-ended tasks, which limits its broader relevance in LLM research.
>
> UQ4CT can be extended to open-ended generation using soft-matching evaluations such as LLM-as-judge or ROUGE. However, for open-ended QA the model produces a sequence, and expert selections evolve across layers × tokens, making functional contributions nested and path-dependent. Simply averaging router weights or using the final token does not capture this structure. A sequence-level FLU formulation is therefore needed, which we leave as future work.
>
> > Theoretical justification is oversimplified. A more rigorous or relaxed analysis would strengthen the paper.
>
> We understand your concern. Our theoretical justification is built on a detailed construction of the functional space presented in Sec. 3.1, where we formally decompose the functional basis induced by MixLoRA and derive how perturbations propagate through the mixture to define functional-level uncertainty. This provides the foundation for the calibration mechanism, and Proposition 3.2 further establishes the optimality of the calibration loss under the true distribution. We also explicitly discuss the gap between theory (global-optimum, infinite-data setting) and practice, and address this through hierarchical decomposition, input-dependent routing, and empirical validation. We have further clarified these theoretical-practical distinctions in the revision.
>
> > Comparable performance to BLoB (N=10). Although the proposed method is more efficient (single-pass), its calibration and accuracy improvements over BLoB(N=10) are modest, suggesting a trade-off between novelty and practical gains.
>
> We thank the reviewer for the comparison. Our method is fundamentally different from BLoB: it provides a single-pass, function-level uncertainty estimate rather than relying on multi-sample posterior draws. This design is the core novelty and yields major efficiency gains. In terms of practical benefits, our approach shows large calibration improvements over N=0 (mean), and achieves performance comparable to BLoB(N=10) while being more sample-efficient.
>
> > Unclear token-level aggregation. In Eq. (12), the aggregation of routing probabilities across tokens is not well defined — it should specify whether it averages over all tokens, the [CLS] token, or the last token.
>
> Since the task is multiple choice, in which the model generates one token as the answer, the aggregation is applied to the last token. We have added a brief explanation to eliminate the ambiguity in the updated version.
>
> > Missing ablation results in main text. The ablation study on the calibration loss weight (Lines 454–458) should be moved from the appendix to the main body for better visibility, since it demonstrates the effectiveness of the calibration component.
>
> Thanks for the suggestion. We moved it to the appendix due to the submission page limitations. We have included it in the main text for the updated version

---

### Official Review · Reviewer_Qqiu · 2025-10-31

**Soundness:** 2
**Presentation:** 2
**Contribution:** 2
**Rating:** 4
**Confidence:** 4

**Summary:**

This paper proposes UQ4CT, which quantifies uncertainty at the functional level during fine-tuning using a Mixture-of-Experts (MoE) architecture with LoRA adapters. The method computes functional-level uncertainty (FLU) from router weights and uses a calibration loss to align FLU with predictive correctness. Experiments on five multiple-choice QA benchmarks show >25% ECE reduction while maintaining accuracy.

**Strengths:**

1. The paper presents a novel calibration mechanism where router weights naturally encode functional-level uncertainty.
2. Achieves calibration during training with only one forward pass at inference, unlike ensemble methods (8× slower) or BLoB(N=10) (10× slower).

**Weaknesses:**

1. All evaluated benchmarks (OBQA, ARC, BOOLQ, ClimateQA, MMLU) are classification tasks. This may limit the method's applicability, as multiple-choice QA is inherently more suitable for calibration compared to open-ended short-form or long-form QA. Can this approach be extended to open-ended QA datasets such as TriviaQA[1]?
2. There is a lack of comparison with alternative uncertainty estimation methods, including self-consistency[2] and training-based approaches[3].
3. Proposition 3.2 only holds at global optimum over true distribution. No analysis of: (a) convergence with non-convex MoE optimization, (b) finite sample effects, (c) conflicts between CE, Lb, and Lcal in Eq. 14. Need empirical verification that learned FLU ≈ P(correct).

[1]TriviaQA: A Large Scale Distantly Supervised Challenge Dataset for Reading Comprehension
[2]Self-Consistency Improves Chain of Thought Reasoning in Language Models
[3]Can AI Assistants Know What They Don't Know?

**Questions:**

1. Line 108: The model name appears inconsistently as "Llama3.1-8B" and "LLaMA3.1-8B." Please standardize the naming convention.
2. What is the performance of the untuned LLaMA3.1-8B on these tasks? This information is necessary to distinguish whether improvements are due to better task learning or improved calibration.
3. In Table~1, it would be clearer to present ECE and ACC on the same row under each dataset for better visual alignment and comparison.

---

> ### Author Response · Authors · 2025-11-24
>
> We appreciate the insightful and constructive review. We address the weaknesses and questions as follows:
>
> > All evaluated benchmarks (OBQA, ARC, BOOLQ, ClimateQA, MMLU) are classification tasks. This may limit the method's applicability, as multiple-choice QA is inherently more suitable for calibration compared to open-ended short-form or long-form QA. Can this approach be extended to open-ended QA datasets such as TriviaQA[1]?
>
> UQ4CT can be extended to open-ended generation using soft-matching evaluations such as LLM-as-judge or ROUGE. However, for open-ended QA the model produces a sequence, and expert selections evolve across layers × tokens, making functional contributions nested and path-dependent. Simply averaging router weights or using the final token does not capture this structure. A sequence-level FLU formulation is therefore needed, which we leave as future work.
>
> > There is a lack of comparison with alternative uncertainty estimation methods, including self-consistency[2] and training-based approaches[3].
>
> We have included the SPUQ(self-consistency) method [1] performance in Appendix A.8 with Llama2-7B. We have updated the results with Llama3.1-8B in the draft. Here is the updated results:
> | Metrics                             | Methods | BoolQ                  | ARC-E                  | ARC-C                  | OBQA                   | ClimateQA              |
> |-------------------------------------|---------|------------------------|------------------------|------------------------|------------------------|------------------------|
> | ACC $\uparrow$ | SPUQ    | $73.79$                | $86.67$                | $73.94$                | $74.00$                | $67.80$                |
> |                                     | UQ4CT   | $89.17_{1.33}$         | $88.66_{0.20}$         | $79.60_{1.21}$         | $88.40_{0.35}$         | $79.97_{0.85}$         |
> | ECE $\downarrow$ | SPUQ    | $7.70$                 | $10.81$                | $7.40$                 | $8.86$                 | $14.68$                |
> |                                     | UQ4CT   | $\mathbf{1.79_{0.43}}$ | $\mathbf{3.97_{0.78}}$ | $\mathbf{4.43_{0.82}}$ | $\mathbf{3.34_{1.60}}$ | $\mathbf{9.36_{2.77}}$ |
>
> [1] Gao, Xiang, et al. "Spuq: Perturbation-based uncertainty quantification for large language models." arXiv preprint arXiv:2403.02509 (2024).
>
> > Proposition 3.2 only holds at global optimum over true distribution. No analysis of: (a) convergence with non-convex MoE optimization, (b) finite sample effects, (c) conflicts between CE, Lb, and Lcal in Eq. 14. Need empirical verification that learned FLU ≈ P(correct).
>
> We appreciate the analysis suggestions. Proposition 3.2 is intended to show that, at global optimum and over the true data distribution, the calibration loss encourages the functional-level uncertainty (FLU) to represent the probability of prediction correctness. We agree that in practical settings, due to non-convexity, finite sample effects, and potential conflicts among CE, Lb, and Lcal in Eq. 14, optimality is not guaranteed. However, our extensive experiments (Tables 1–3) empirically verify that the learned FLU is well-calibrated (i.e., closely tracks empirical accuracy) and our method consistently reduces calibration error without sacrificing accuracy. We will clarify these theoretical and practical distinctions in the revision.

---

> > ### Author Response · Authors · 2025-11-24
> >
> > > Line 108: The model name appears inconsistently as "Llama3.1-8B" and "LLaMA3.1-8B." Please standardize the naming convention.
> >
> > Thanks for the correction. We have included this in the updated version.
> >
> > > What is the performance of the untuned LLaMA3.1-8B on these tasks? This information is necessary to distinguish whether improvements are due to better task learning or improved calibration.
> >
> > We have included the base model comparison in the updated version. As shown in the following table, our proposed method achieves improved calibration across all tasks, while maintaining consistent accuracy improvements as other fine-tuning methods.
> >
> > | Metrics                   | Methods                      |            BoolQ           |            ARC-E           |            ARC-C           |            OBQA            |          ClimateQA         |
> > |---------------------------|------------------------------|:--------------------------:|:--------------------------:|:--------------------------:|:--------------------------:|:--------------------------:|
> > | ACC $\uparrow$   | Base Model |  $74.73$ |  $87.27$ |  $74.32$ |  $72.80$ |  $68.64$ |
> > |                           | LoRA                         | $\underline{89.73_{0.58}}$ | $\underline{88.82_{1.82}}$ |       $78.21_{0.73}$       |       $88.00_{1.22}$       |       $78.25_{1.29}$       |
> > |                           | MC Drop                      |       $89.65_{0.55}$       |       $88.16_{1.75}$       |       $77.14_{0.69}$       |       $87.12_{1.18}$       |       $78.19_{1.35}$       |
> > |                           | Ensemble                     |   $\mathbf{89.87_{0.43}}$  |   $\mathbf{89.14_{1.31}}$  |       $78.81_{0.96}$       |       $86.47_{0.42}$       |       $78.53_{2.98}$       |
> > |                           | MixLoRA                      |       $88.68_{0.82}$       |       $87.74_{0.36}$       |       $78.56_{1.87}$       | $\underline{88.27_{0.50}}$ | $\underline{79.94_{1.29}}$ |
> > |                           | LA                           |       $89.58_{0.19}$       |       $86.22_{3.52}$       |       $78.00_{3.76}$       |       $86.00_{6.01}$       |       $79.82_{3.48}$       |
> > |                           | BLoB(Mean)                   |       $89.02_{0.93}$       |       $88.71_{0.82}$       |       $79.37_{0.71}$       |       $87.60_{1.04}$       |       $79.02_{0.50}$       |
> > |                           | BLoB(N=10)                   |       $89.39_{1.13}$       |       $87.96_{0.62}$       |   $\mathbf{80.08_{1.55}}$  |       $87.13_{0.88}$       |       $79.02_{0.50}$       |
> > |                           | UQ4CT                        |       $89.17_{1.33}$       |       $88.66_{0.20}$       | $\underline{79.60_{1.21}}$ |   $\mathbf{88.40_{0.35}}$  |   $\mathbf{79.97_{0.85}}$  |
> > | ECE $\downarrow$ | Base Model |  $6.94$  |  $13.76$ |  $11.30$ |  $11.39$ |  $14.58$ |
> > |                           | LoRA                         |       $15.82_{0.57}$       |        $6.55_{1.70}$       |       $14.07_{0.68}$       |        $7.30_{0.43}$       |       $13.70_{1.50}$       |
> > |                           | MC Drop                      |       $14.73_{0.54}$       |        $6.48_{1.74}$       |       $14.12_{0.71}$       |        $7.24_{0.39}$       |       $13.11_{1.46}$       |
> > |                           | Ensemble                     |       $14.56_{0.55}$       |        $7.08_{0.73}$       |       $13.71_{1.29}$       |        $8.63_{0.38}$       |       $14.69_{0.84}$       |
> > |                           | MixLoRA                      |       $15.85_{0.76}$       |        $7.79_{0.45}$       |       $13.71_{1.90}$       |        $6.58_{0.21}$       |       $14.68_{0.09}$       |
> > |                           | LA                           |        $3.78_{0.60}$       |        $7.63_{1.71}$       |        $8.92_{4.16}$       |       $11.97_{5.97}$       |       $11.48_{1.66}$       |
> > |                           | BLoB(Mean)                   |        $7.54_{0.57}$       |        $4.89_{0.32}$       |       $11.26_{1.13}$       |        $6.83_{0.90}$       |       $12.74_{0.88}$       |
> > |                           | BLoB(N=10)                   |  $\underline{2.76_{0.41}}$ |   $\mathbf{3.35_{0.50}}$   |  $\underline{6.81_{1.43}}$ |  $\underline{3.84_{1.00}}$ | $\underline{11.96_{2.57}}$ |
> > |                           | UQ4CT                        |   $\mathbf{1.79_{0.43}}$   |  $\underline{3.97_{0.78}}$ |   $\mathbf{4.43_{0.82}}$   |   $\mathbf{3.34_{1.60}}$   |   $\mathbf{9.36_{2.77}}$   |
> >
> > > In Table~1, it would be clearer to present ECE and ACC on the same row under each dataset for better visual alignment and comparison.
> >
> > We appreciate the suggestion. However, we prefer to keep ECE and ACC on separate rows, as they are distinct metrics with opposite directions for optimality (higher is better for ACC, lower is better for ECE). This separation provides a clearer comparison.

---

> > > ### Comment · Reviewer_Qqiu · 2025-11-26
> > >
> > > Thanks for the rebuttal. I will keep my score as my concerns are only partially addressed.

---

### Author Response · Authors · 2025-12-03
**Overall Response**

We thank all reviewers for their thoughtful evaluations and constructive feedback on our submission. We would like to emphasize several points clarified in our revision:

1. **Methodological Generality**: We demonstrate the generality of UQ4CT by including results on both Llama-3.1-8B (main text) and Mistral-7B (Appendix A.3). Across both model families, UQ4CT consistently outperforms baseline methods by delivering substantially improved calibration while maintaining competitive accuracy, including under distribution shift.

2. **Comparisons and Scope**: We added new baselines (self-consistency method applied on Llama3.1-8B, namely SPUQ) and clarified that our approach is complementary to both Bayesian and sampling-based UQ frameworks. Across all benchmarks, UQ4CT achieves superior calibration with only a single forward pass, matching or exceeding the accuracy of existing methods.


| Metrics                             | Methods | BoolQ                  | ARC-E                  | ARC-C                  | OBQA                   | ClimateQA              |
|-------------------------------------|---------|------------------------|------------------------|------------------------|------------------------|------------------------|
| ACC $\uparrow$   | SPUQ    | $73.79$                | $86.67$                | $73.94$                | $74.00$                | $67.80$                |
|                                     | UQ4CT   | $89.17_{1.33}$         | $88.66_{0.20}$         | $79.60_{1.21}$         | $88.40_{0.35}$         | $79.97_{0.85}$         |
| ECE $\downarrow$ | SPUQ    | $7.70$                 | $10.81$                | $7.40$                 | $8.86$                 | $14.68$                |
|                                     | UQ4CT   | $\mathbf{1.79_{0.43}}$ | $\mathbf{3.97_{0.78}}$ | $\mathbf{4.43_{0.82}}$ | $\mathbf{3.34_{1.60}}$ | $\mathbf{9.36_{2.77}}$ |




3. **Base Model Performance**: Following reviewer requests, we now include base model performance for all evaluated tasks. These comparisons further highlight that UQ4CT improves calibration well beyond both base and fine-tuned models, while preserving or improving predictive accuracy.

4. **Theoretical Clarification**: We clarified the naming and interpretation of our functional-level metric (renaming FLU to FLC for accuracy) and explicitly discussed the limitations of the theoretical justification (global optimum vs. practical non-convexity), supported by comprehensive empirical verification.

5. **Ablation and Analysis**: At the reviewers’ request, we moved ablation results for the calibration loss into the main text, providing clearer evidence of the effectiveness of our calibration term across settings. We also provided detailed answers to all technical questions regarding routing aggregation, computational complexity, and baseline comparisons.

6. **Limitation and Future Directions**: We acknowledge UQ4CT’s limitation to tasks with well-defined correctness signals, and explicitly outline directions for extending sequence-level FLC to open-ended generation.

We hope these updates and clarifications address the reviewers’ main concerns and demonstrate the value and rigor of UQ4CT for the community.

---

> ### Author Response · Authors · 2025-12-03
> **Performance Comparison with Base Model**
>
> | Metrics                   | Methods                      |            BoolQ           |            ARC-E           |            ARC-C           |            OBQA            |          ClimateQA         |
> |---------------------------|------------------------------|:--------------------------:|:--------------------------:|:--------------------------:|:--------------------------:|:--------------------------:|
> | ACC $\uparrow$   | Base Model |  $74.73$ |  $87.27$ |  $74.32$ |  $72.80$ |  $68.64$ |
> |                           | LoRA                         | $\underline{89.73_{0.58}}$ | $\underline{88.82_{1.82}}$ |       $78.21_{0.73}$       |       $88.00_{1.22}$       |       $78.25_{1.29}$       |
> |                           | MC Drop                      |       $89.65_{0.55}$       |       $88.16_{1.75}$       |       $77.14_{0.69}$       |       $87.12_{1.18}$       |       $78.19_{1.35}$       |
> |                           | Ensemble                     |   $\mathbf{89.87_{0.43}}$  |   $\mathbf{89.14_{1.31}}$  |       $78.81_{0.96}$       |       $86.47_{0.42}$       |       $78.53_{2.98}$       |
> |                           | MixLoRA                      |       $88.68_{0.82}$       |       $87.74_{0.36}$       |       $78.56_{1.87}$       | $\underline{88.27_{0.50}}$ | $\underline{79.94_{1.29}}$ |
> |                           | LA                           |       $89.58_{0.19}$       |       $86.22_{3.52}$       |       $78.00_{3.76}$       |       $86.00_{6.01}$       |       $79.82_{3.48}$       |
> |                           | BLoB(Mean)                   |       $89.02_{0.93}$       |       $88.71_{0.82}$       |       $79.37_{0.71}$       |       $87.60_{1.04}$       |       $79.02_{0.50}$       |
> |                           | BLoB(N=10)                   |       $89.39_{1.13}$       |       $87.96_{0.62}$       |   $\mathbf{80.08_{1.55}}$  |       $87.13_{0.88}$       |       $79.02_{0.50}$       |
> |                           | UQ4CT                        |       $89.17_{1.33}$       |       $88.66_{0.20}$       | $\underline{79.60_{1.21}}$ |   $\mathbf{88.40_{0.35}}$  |   $\mathbf{79.97_{0.85}}$  |
> | ECE $\downarrow$ | Base Model |  $6.94$  |  $13.76$ |  $11.30$ |  $11.39$ |  $14.58$ |
> |                           | LoRA                         |       $15.82_{0.57}$       |        $6.55_{1.70}$       |       $14.07_{0.68}$       |        $7.30_{0.43}$       |       $13.70_{1.50}$       |
> |                           | MC Drop                      |       $14.73_{0.54}$       |        $6.48_{1.74}$       |       $14.12_{0.71}$       |        $7.24_{0.39}$       |       $13.11_{1.46}$       |
> |                           | Ensemble                     |       $14.56_{0.55}$       |        $7.08_{0.73}$       |       $13.71_{1.29}$       |        $8.63_{0.38}$       |       $14.69_{0.84}$       |
> |                           | MixLoRA                      |       $15.85_{0.76}$       |        $7.79_{0.45}$       |       $13.71_{1.90}$       |        $6.58_{0.21}$       |       $14.68_{0.09}$       |
> |                           | LA                           |        $3.78_{0.60}$       |        $7.63_{1.71}$       |        $8.92_{4.16}$       |       $11.97_{5.97}$       |       $11.48_{1.66}$       |
> |                           | BLoB(Mean)                   |        $7.54_{0.57}$       |        $4.89_{0.32}$       |       $11.26_{1.13}$       |        $6.83_{0.90}$       |       $12.74_{0.88}$       |
> |                           | BLoB(N=10)                   |  $\underline{2.76_{0.41}}$ |   $\mathbf{3.35_{0.50}}$   |  $\underline{6.81_{1.43}}$ |  $\underline{3.84_{1.00}}$ | $\underline{11.96_{2.57}}$ |
> |                           | UQ4CT                        |   $\mathbf{1.79_{0.43}}$   |  $\underline{3.97_{0.78}}$ |   $\mathbf{4.43_{0.82}}$   |   $\mathbf{3.34_{1.60}}$   |   $\mathbf{9.36_{2.77}}$   |

---

### Meta-Review · Area_Chair_7ocf · 2026-01-07

**Summary:**

This paper proposes a fine-tuning framework for uncertainty quantification in LLMs with MoE LoRA experts. The method involves a functional-level confidence signal from MoE routing probabilities and introduces a calibration loss to align this signal with prediction correctness during training.

In general, reviewers hold conservative opinions toward the paper. Reviewers acknowledge the novelty of leveraging MoE routing to model functional-level uncertainty (Qqiu, 6Wcn, yiFV), the efficiency of achieving calibration with a single forward pass (Qqiu, 6Wcn), and the generally strong empirical results (d28n). However, reviewers also raise several concerns. The major concerns are as follows:
- Scope and generalizability. Multiple reviewers question whether the proposed approach generalizes beyond the current experimental setting (Qqiu, 6Wcn). Concerns include the focus on discriminative, multiple-choice tasks, reliance on Mixture-of-Experts architectures.
- Theoretical justifications. Reviewers raise concerns that the theoretical analysis relies on simplified analysis that may not hold in practical training settings (Qqiu, 6Wcn).
- Baseline comparisons and advantage. Reviewers note missing comparisons with some uncertainty estimation baselines and find that improvements over strong methods such as BLoB(N=10) are modest (Qqiu, 6Wcn). They also point out that the computational cost comparison is not entirely clear due to architectural differences, weakening the empirical advantage of the method (d28n).

**Reviewer Concerns:**

The authors made a substantial effort during the rebuttal phase and addressed a number of concerns. In particular, they responded to the issues of baseline comparisons raised by Qqiu, the concerns about performance gains raised by 6Wcn, and provided reasonable clarification regarding the theoretical analysis. However, concerns about the scope and generalization ability of the proposed approach remain. Specifically, reviewers note that the method is evaluated only on discriminative tasks, rather than generative settings such as open-ended QA. While the authors frame this limitation as future work, the concern remains unresolved, which could be a major limitation.

**Reviewer Scores:**

Reviewer Qqiu (rating: 4, confidence: 4): The reviewer explicitly states that they will keep the score unchanged, as the concerns are only partially addressed.

Reviewer 6Wcn (rating: 4, confidence: 3): This reviewer did not participate further in the discussion. However, similar to Qqiu, the concern regarding limited application scope remains unresolved, so the score is likely to remain at 4.

Reviewer d28n and yiFV: These reviewers did not participate further in the discussion. Based on the rebuttal, their major technical and clarification concerns are generally addressed. However, depending on how strongly they weigh the remaining scope and generalization issues raised by Qqiu and 6Wcn, their scores are likely to remain at 4 or increase modestly to 6.

---

### Decision · Program_Chairs · 2026-01-26

Reject